# Adapting Language Models to Compress Contexts

**Alexis Chevalier**[*]    **Alexander Wettig**[*]    **Anirudh Ajith**    **Danqi Chen**

Department of Computer Science & Princeton Language and Intelligence

Princeton University

{achevalier,anirudh.ajith}@princeton.edu

{awettig, danqic}@cs.princeton.edu

## Abstract

Transformer-based language models (LMs) are powerful and widely-applicable tools, but their usefulness is constrained by a finite context window and the expensive computational cost of processing long text documents. We propose to adapt pre-trained LMs into *AutoCompressors*. These language models are capable of compressing long contexts into compact *summary vectors*, which are then accessible to the model as soft prompts. Summary vectors are trained with an unsupervised objective, whereby long documents are processed in segments, and summary vectors from all previous segments are used in language modeling. We fine-tune OPT and Llama-2 models on sequences of up to 30,720 tokens and show that AutoCompressors can utilize long contexts to improve perplexity. We evaluate AutoCompressors on in-context learning by compressing task demonstrations and find that summary vectors are good substitutes for plain-text demonstrations, increasing accuracy while reducing inference costs. Finally, we explore the benefits of pre-computing summary vectors for large corpora by applying summary vectors to retrieval-augmented language modeling and a passage re-ranking task. Overall, AutoCompressors emerge as a simple and inexpensive solution to extend the context window of LMs while speeding up inference over long contexts.[1]

## 1 Introduction

Transformer-based ([Vaswani et al., 2017](#)) language models (LMs) have recently seen a sharp rise in popularity and are now receiving millions of queries, processing billions of tokens, and generating text for a wide variety of applications ([Brown et al., 2020](#); [Touvron et al., 2023](#); [Zhang et al.,

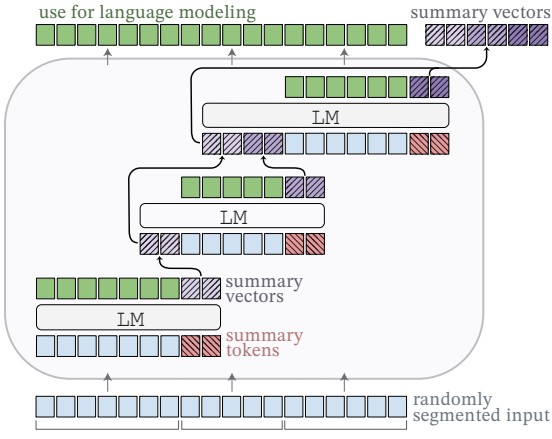

Figure 1: *AutoCompressors* process long documents by recursively generating summary vectors which are passed as soft prompts to all subsequent segments.

2022](#)). With this rise in popularity comes the challenge for researchers to make LMs more *efficient*, to speed up inference and to deploy LMs at scale, while increasing their *versatility*, thus allowing users to process more data in new ways.

With these goals in mind, we propose to teach pre-trained LMs the ability to compress text into *summary vectors*. Summary vectors are short soft prompts ([Lester et al., 2021](#)), one or two orders of magnitude shorter than the pre-compressed plain text, that are obtained from the output states of a language model. Summary vectors serve two general purposes: they can help extend the language model's context window to very long documents with minimal computational overhead, and they help speed up inference on text for which summary vectors have been pre-computed and cached.

Our models, which we call AutoCompressors, are trained with a simple unsupervised learning objective that encourages the model to store essential information in the summary vectors. Summary vectors are produced segment by segment from long documents and are used to improve language modeling in future segments (Figure 1). Our work

---

[*]AC and AW contributed equally. This work was done when AC was at the Institute for Advanced Study and visited the Princeton NLP group.

[1]Our code and models are publicly available at https://github.com/princeton-nlp/AutoCompressors.

builds on the recently proposed RMT architecture (Bulatov et al., 2022) with a crucial difference: we introduce *summary accumulation*, in which summary vectors from all segments are concatenated to produce the summary of the entire document. We also train AutoCompressors with randomly segmented inputs so they can better compress contexts of variable lengths in downstream tasks. We show that these innovations improve long-range information retention and enable new ways of reasoning over multiple passages.

AutoCompressors can be initialized with pre-trained LMs to produce powerful and versatile models. We fine-tune AutoCompressors from OPT-2.7B (Zhang et al., 2022) and Llama-2-7B (Touvron et al., 2023) models on sequences from 6,144 up to 30,720 tokens with a single NVIDIA A100 GPU of 80GB memory. We show that summary vectors are effective for improving perplexity over long documents and that these compression capabilities are robust to domain generalization. Our analysis suggests that AutoCompressors are able to reason over summary vectors, making them useful for a diverse set of downstream applications.

We apply AutoCompressors to in-context learning (ICL) by compressing up to 90 in-context demonstrations. We consider 11 classification tasks, including 7 SuperGLUE tasks (Wang et al., 2019), and we find that summary vectors outperform few-shot ICL with a comparable number of in-context tokens on 8 out of 11 tasks.

Finally, we explore two applications where AutoCompressors can reduce inference costs by pre-computing summary vectors for large corpora. First, we adopt a setting for retrieval-augmented language modeling (Shi et al., 2023). We find that for equal sequence lengths, using summary vectors achieves $1.5\times$ the perplexity gains compared to plain-text passages, and outperforms retrieval-augmented methods for similar computational budgets. Secondly, we consider a zero-shot passage re-ranking task (Sachan et al., 2022). We establish that re-ranking passages based on their summary vectors achieves the best trade-off between re-ranking performance and inference throughput.

In summary, our main contributions are the following: (1) We introduce a method for extending LMs to long context windows under small-scale computational requirements by learning to generate summary vectors. We propose summary accumulation and training with randomized segmenting as

key features of AutoCompressors. (2) We show that summary vectors encode useful information for downstream tasks and can be used to reduce the inference cost of in-context learning. (3) We demonstrate the benefits of pre-computing summary vectors for large corpora and using AutoCompressors in conjunction with retrievers.

## 2   Related Work

**Soft prompts**  Soft prompt tuning is an effective method to adapt pre-trained Transformers without updating existing parameters (Lester et al., 2021; Zhong et al., 2021; Liu et al., 2022). Newly initialized embeddings are prepended to the input sequence (the "soft prompt"), and optimization is performed with respect to these new parameters while the rest of the model is frozen. It is one of many parameter-efficient fine-tuning methods (Lialin et al., 2023) and is related to prefix tuning, where newly initialized parameters are prepended to the attention states instead (Li and Liang, 2021).

**Prompt compression**  Wingate et al. (2022) propose to learn a soft prompt $\sigma$ to compress the information contained in a context $x$. Given a pre-trained language model $p_{\mathrm{LM}}$, they draw continuations $y \sim p_{\mathrm{LM}}(\cdot \mid x)$ based on $x$ and use a distillation objective to align the model's predictions conditioned on the soft prompt $p_{\mathrm{LM}}(y \mid \sigma)$ to the predictions conditioned on the context $p_{\mathrm{LM}}(y \mid x)$. Wingate et al. (2022) find that soft prompts retain high-level information and facilitate controllable generation. However, the approach requires running the optimization for every new context $x$, with no knowledge transfer between similar contexts. In contrast, our AutoCompressors learn to predict their own soft prompts $\sigma$ as a function of $x$.

**Context distillation**  A related line of work (Askell et al., 2021; Snell et al., 2022) aims to distill in-context information, e.g., instructions, into an unprompted student model. In concurrent work, Mu et al. (2023) teach models to compress instructions into short key-value attention prefixes. Our approach differs by learning to compress any context information, including long documents, and results in more compact soft prompts.

**Long-range Transformers**  A number of architectural modifications have been proposed to scale Transformers to longer context lengths while reducing the high memory costs of full attention. These include restricting and sparsifying the attention window (Dai et al., 2019; Child et al., 2019), ap-

proximating the attention (Rae et al., 2020; Zheng et al., 2022; Choromanski et al., 2021), as well as introducing recurrent elements (Ma et al., 2022; Bulatov et al., 2022), conditional computation (Ainslie et al., 2023), and retrieving previous tokens from the context at the output layer (Zhong et al., 2022). See Tay et al. (2022) for a comprehensive survey of efficient long-range architectures.

Most of these architectures typically require expensive training from scratch, or will deviate substantially from a pre-trained initialization.[2] Moreover, many language models lack the inductive bias to extrapolate to longer sequences (Press et al., 2022). While AutoCompressors could in principle be trained from scratch, we show that they offer a straightforward solution for extending the context window of pre-trained models to longer sequences.

## 3 Method

We describe how we adapt a pre-trained language model to compress text into summary vectors. An overview of our architecture is shown in Figure 1.

**Summary vectors** The AutoCompressor builds on the RMT architecture (Bulatov et al., 2022). We extend the input vocabulary of the base model by $\kappa$ special summary tokens <Sum>$_i$ and initialize $\kappa$ new input embeddings.[3] When we append the sequence <Sum>$_1$ . . . <Sum>$_\kappa$ to an input, it signals to the model to output special *summary vectors* of the preceding context. These vectors can then be passed to the next text segment as a soft prompt of length $\kappa$. Since the embedding spaces of pre-trained language models can span thousands of dimensions, we expect that this mechanism has a high capacity for passing information to subsequent segments. Furthermore, a soft prompt can interpolate between many token embeddings, and therefore represent more abstract concepts than a single discrete token (Wingate et al., 2022).

**Summary accumulation** We split long documents into segments $S_1, \ldots, S_n$ and process them sequentially. Bulatov et al. (2022) incorporate information from previous segments by prepending the compressed summary $\sigma_{i-1}$ produced from $S_{i-1}$ to the embedded inputs of $S_i$. We propose *summary accumulation*, which allows for a direct information pathway between each segment and all segments preceding it: we concatenate the summary vectors $\sigma_1 \ldots, \sigma_{i-1}$ to form $\sigma_{<i}$ and prepend $\sigma_{<i}$ to $S_i$. Note that the length of $\sigma_{<i}$ is now $(i-1)\kappa$, which grows linearly with the document length.

**Positional embeddings** When using a base Transformer architecture with absolute positional embeddings, such as the OPT architecture (Zhang et al., 2022), we do not add positional embeddings to the summary tokens <Sum>$_i$, nor to the summary vectors. This allows us to use all pre-trained position embeddings as context tokens and makes it possible to scale the model to an arbitrary number of compression steps during training. The model still preserves the order of summary tokens due to their separate token embeddings.

If the base Transformer uses relative positional embeddings, such as RoPE (Su et al., 2022), we apply the positional embedding to the summary tokens and vectors without any further modification.

### 3.1 Training Summary Vectors

We use a simple unsupervised training approach which encourages the model to learn to compress contexts over multiple steps.

**Training objective** Write $(x_1^i, \ldots, x_{m_i}^i)$ for the segment $S_i$ for every $i \leq n$, where $m_i$ is the number of tokens in $S_i$. Conditioning on the concatenated summary vectors $\sigma_{<i}$, we project the Transformer outputs with the language modeling head to obtain the next-token probabilities $p(x_t^i \mid x_1^i, \ldots, x_{t-1}^i, \sigma_{<i})$. We minimize the cross-entropy loss over the entire document:

$$\mathcal{L} = -\frac{1}{N} \sum_{i=1}^{n} \sum_{t=1}^{m_i} \log p(x_t^i \mid x_1^i, \ldots, x_{t-1}^i, \sigma_{<i}).$$

where $N$ is the total number of tokens. This objective retains the pre-trained language model's abilities on the first segment $S_1$ and it incentivizes the model to store useful information in the summary vectors, which future segments can leverage to make better token predictions.

Unlike Wingate et al. (2022), we do not train with a knowledge distillation objective, since the pre-trained LM has a limited context window as a teacher, whereas the AutoCompressor student learns to process much longer documents.

**Randomized segmenting** We randomly vary the lengths $m_i$ of the segments $S_i$ during training, subject to the condition that each segment fits into

---

[2]In our pre-liminary experiments, even fine-tuning a pre-trained OPT-2.7b model with Transformer-XL-style training (Dai et al., 2019) caused optimization difficulties and deteriorated the pre-trained model quality.

[3]When fine-tuning OPT models, we observe benefits with initializing the embeddings of the summary tokens with the pre-trained embedding for the end-of-sequence token .

the model's context window. This allows Auto-Compressors to compress documents of different lengths and improves performance under evaluation with fixed-length segments (see Figure 2).

**BPTT with stop-gradients** We employ backpropagation through time (BPTT) and gradient checkpointing (Chen et al., 2016) for each segment to reduce the size of the computational graph. In addition, we compute and cache summary vectors and stop their gradients after 2 compression steps, similar to caching past attention states in Transformer-XL training (Dai et al., 2019). This assumes that for learning to compress the useful information in $S_i$, it is sufficient to predict the tokens in the adjacent $S_{i+1}$. In Figure 2, we confirm that this incurs no penalty when predicting long segments, while further reducing GPU memory requirements.

## 4 Language Modeling Evaluation

In this section, we train AutoCompressors and evaluate their long-range language modeling capabilities by sampling long sequences which we split into segments of 2,048 tokens. We fix the final segment and compress the previous $n$ segments. We track the perplexity of the final segment when conditioning on the summary vectors for each $n$.

We conduct our main experiments and ablations with OPT models (Zhang et al., 2022) of 1.3B or 2.7B parameters, fine-tuned on 2B tokens from the Pile (Gao et al., 2020). In Section 4.1, we evaluate an AutoCompressor on sequences of 8,000 tokens and compare to an equivalent RMT model and an Extended Full Attention baseline. In Section 4.2, we fine-tune an AutoCompressor on sequences of 30,000 tokens to demonstrate the feasibility on very long sequences. Finally, in Section 4.3, we scale up AutoCompressors by fine-tuning a Llama-2-7B model on 15B tokens from RedPajama (TogetherAI, 2023). Full model hyperparameters and data information can be found in Appendix A.

### 4.1 Experiments on 8K-Token Sequences

**Setting** We initialize all models with the 2.7B-parameter OPT model and fine-tune on 2B tokens from 4 domains form the Pile (Gao et al., 2020). Our AutoCompressor uses $\kappa = 50$ summary tokens and is fine-tuned with summary accumulation over four segments, each ranging from 1,024 to 2,048 tokens. Compressing 2,048 tokens into 50 summary vectors achieves a compression rate of 40 tokens per summary vector. We use the following

baselines:
1. We fine-tune an OPT-2.7B baseline on our data. This model is limited to sequences of 2,048 tokens due to pre-training.
2. Extended full attention: We fine-tune OPT-2.7B on sequences of up to 4,096 tokens by extending the model's positional embeddings. We initialize the embeddings for positions $[2049..4096]$ with the embeddings for positions $[1..2048]$. We are not able to extend the context beyond 4,096 tokens due to GPU memory limitations.
3. RMT-2.7B: We fine-tune an RMT model on our data with $\kappa = 50$ summary vectors.

We evaluate on documents of 8,192 tokens, drawn from the 4 training domains or 4 held-out domains. We generate summary vectors for up to 3 segments of 2,048 tokens, but also for single segments as short as 128 tokens. For the extended full-attention baseline we prepend the previous context tokens to the context window.

**Results** We show the results in Table 1. We find that the AutoCompressor benefits from long contexts of 6,144 tokens and consistently outperforms the RMT model.

We also find that the AutoCompressor benefits from much shorter sequences than seen during training, unlike RMT. See also Figure 2 and Table 6 for the usefulness of randomized segmenting.

While extended full attention performs the best on 4,096-long sequences, we observe a trade-off for shorter contexts where AutoCompressors achieve the best performance. We also stress that the AutoCompressor attends to at most 150 additional soft prompts during evaluation, whereas the full attention model is given an additional 2,048 tokens.

These trends hold for both in-domain and out-of-domain evaluation. However, the gap between the AutoCompressor and the full-attention baseline increases in the out-of-domain setting, suggesting that the summary vectors generalize slightly less than pre-trained attention heads.

### 4.2 Experiments on 30K-Token Sequences

**Setting** We fine-tune OPT-1.3B and OPT-2.7B as AutoCompressors on 2B tokens but train on sequences of 30,720 tokens with 20 compression steps.[4] We use 50 summary tokens, randomized segmenting, and stop-gradients as before. We also

---

[4] Due to the scarcity of very long sequences in the Pile, we only train on data from the Books3 domain, and use the Gutenberg domain as out-of-domain evaluation.

| | In-domain | | | | | Out-of-domain | | | | |
|---|---|---|---|---|---|---|---|---|---|---|
| *Segments* | | *1* | | – 2 – | – 3 – | | *1* | | – 2 – | – 3 – |
| Context tokens | 128 | 512 | 2048 | 4096 | 6144 | 128 | 512 | 2048 | 4096 | 6144 |
| Extended FA† | 6.33† ↑1.0% | 6.15† ↓2.1% | 5.94† ↓5.4% | - | - | 8.57† ↑0.5% | 8.28† ↓2.9% | 7.93† ↓7.0% | - | - |
| RMT | 6.42 ↑2.2% | 6.19 ↓1.4% | 6.02 ↓4.1% | 6.02 ↓4.1% | 6.01 ↓4.3% | 8.76 ↑2.7% | 8.44 ↓1.1% | 8.21 ↓3.8% | 8.20 ↓3.9% | 8.20 ↓3.9% |
| AutoCompressor | 6.14 ↓2.2% | 6.04 ↓3.8% | 5.98 ↓4.8% | 5.94 ↓5.4% | **5.93** ↓5.6% | 8.39 ↓1.6% | 8.26 ↓3.2% | 8.17 ↓4.2% | 8.12 ↓4.8% | **8.10** ↓5.0% |

Table 1: Held-out perplexity on 2,048 tokens, while varying the length of the preceding context (all the experiments are based on OPT-2.7B models). For RMT and AutoCompressor, we condition on summary vectors. We also report the perplexity gains compared to the fine-tuned OPT baseline without extra context, which achieves 6.28 in-domain and 8.53 out-of-domain (gains shown in colored numbers). †: Although the extended full attention (Extended FA) achieves similar or slightly better perplexity, it uses up to 2,048 additional tokens and cannot extend further. However, the AutoCompressor uses only $50 \times 3 = 150$ summary vectors to process 6,144 context tokens.

| *Segments* | – 0 – | – 7 – | – 14 – | CUDA |
|---|---|---|---|---|
| Context tokens | 0 | 14336 | 28672 | memory |
| RMT-1.3B | 13.18 | 12.50 | 12.50 | 54GB |
| AutoCompressor-1.3B | 13.21 | 12.49 | **12.47** | 38GB |
| RMT-2.7B | - | - | - | OOM |
| AutoCompressor-2.7B | 11.86 | 11.21 | **11.18** | 75GB |

Table 2: Evaluation results for AutoCompressors trained on sequences of 30,720 tokens and evaluated on Books3 (in-domain) and Gutenberg (out-of-domain). We train with a single NVIDIA A100 GPU and report the CUDA memory required for fine-tuning using a single sequence per batch. AutoCompressors require less memory because we stop gradients after two segments.

| *Segments* | – 0 – | | *1* | | – 2 – | – 3 – |
|---|---|---|---|---|---|---|
| Context tokens | 0 | 128 | 512 | 2048 | 4096 | 6144 |
| Llama-2 | 5.52 | 5.30 | 5.15 | 4.98 | - | - |
| Extended FA | 5.40 | 5.19 | 5.06 | 4.88 | 4.80 | 4.76 |
| AutoCompressor | 5.40 | 5.23 | 5.16 | 5.11 | 5.08 | **5.07** |

Table 3: Evaluation results for our AutoCompressor trained from Llama-2 7B on sequences of 6,144 tokens. For the AutoCompressor, we condition on summary vectors. For Llama-2 and the Extended Full Attention (Extended FA), we condition on plain text tokens.

fine-tune an RMT model from OPT-1.3B, to use as a baseline. We are not able to fine-tune a 2.7B-parameter RMT baseline because the RMT method leads to an out-of-memory error.

All models are evaluated on the final 2,048 held-out tokens of documents of size 30,720 tokens by compressing all previous 2,048-token segments.

**Results** We collect our results in Table 2. The evaluation shows that both AutoCompressor models learn to utilize the entire 28K tokens to reduce perplexity, while the RMT baseline does not benefit from doubling the number of context tokens from 14K to 28K. This shows that summary accumula-

tion effectively captures long-range dependencies in documents. We also report the CUDA memory requirements for fine-tuning each model in Table 2. We train with one NVIDIA A100 GPU with 80GB of memory. Stopping gradients reduces CUDA memory and makes it possible to fine-tune an AutoCompressor from OPT-2.7B, while fine-tuning with RMT leads to out-of-memory at that scale.

### 4.3 Scaling Up AutoCompressors to Llama-2

**Setting** We fine-tune a 7B-parameter Llama-2 model as an AutoCompressor on a single GPU by freezing the model and optimizing only the summary token embeddings and the attention weights via LoRA (Hu et al., 2022). The model is trained on 15B tokens from RedPajama (TogetherAI, 2023), split into sequences of 6,144 tokens, and we use 50 summary tokens, randomized segmenting, and stop-gradients. We also fine-tune an Extended Full Attention baseline on the same dataset. The context window of the pre-trained model is extended by increasing the $\theta$ value in RoPE following (Rozière et al., 2023).

We compare both models to the pre-trained Llama-2-7B model, which has a context window of 4,096 tokens. All models are evaluated on the final 2,048 tokens of 8,192-token documents.

**Results** We collect our results in Table 3. The AutoCompressor benefits from the entire context to reduce perplexity: compressing a 4,096-token context into 100 summary vectors achieves similar perplexity to the Extended Full Attention baseline with 512 plain text tokens, and compressing a 6,144-token context into 150 summary vectors further improves perplexity slightly. Moreover, we find that summary vectors preserve perplexity when short contexts are compressed.

However, Llama-2 and the Extended Full At-

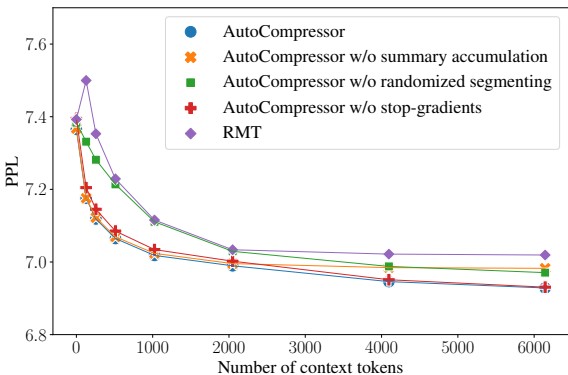

Figure 2: Perplexity on 2048 held-out tokens given different numbers of compressed tokens. Compression is performed on up to 3 segments of 2048 tokens. Ablations show that the different components of our fine-tuning strategy help boost performance and that stop-gradients do not impact performance.

tention baseline outperform the AutoCompressor when longer contexts are provided. Further research is needed to construct summary vectors that preserve all of the context information.

### 4.4 Analysis

**Ablations** We train OPT-2.7B models without randomized segmenting, summary accumulation, or stop gradients. The results are shown in Figure 2. We find that randomized segmenting leads to better compression of short segments, but still improves perplexity when compressing multiple 2048 token segments. As expected, summary accumulation helps improve perplexity beyond one compressed segment. We also confirm that stopping gradients does not impact performance despite reducing GPU memory requirements. In Table 2, we also show that stopping gradients helps reduce GPU memory.

We also train AutoCompressors with $\kappa = 20$, 50, 70 or 100 summary tokens and report the held-out perplexity results in Table 7 in the Appendix. Surprisingly, we find that performance does not increase with longer soft prompts, and $\kappa = 50$ performs the best overall. We hypothesize that learning a larger number of summary vectors may require a larger training budget.

**Token-level analysis** We seek to better understand how summary vectors benefit individual token predictions. In Figure 5 in the Appendix, we show perplexity gains at each token position for the AutoCompressor with summary vectors and for the extended full-attention baseline.

We find that conditioning on summary vectors

improves perplexity over all 2048 token positions. We observe that the extended full attention baseline outperforms the AutoCompressor at the start of the sequence, whereas the AutoCompressor achieves the best performance towards the end of the sequence. This shows that summary vectors effectively capture long-range textual dependencies.

In Appendix D, we show examples of sentences and tokens which benefit the most from summary vectors. We find that summary vectors contain salient information, such as names or dates, and that the model can reason over summary vectors. This confirms that summary vectors are useful summaries of the compressed text.

## 5 Compressing Demonstrations for In-Context Learning

In this section, we study the usefulness of summary vectors for performing downstream tasks. We show that in-context demonstrations can reliably be compressed down into summary vectors to improve performance while also increasing efficiency on a diverse set of NLP benchmarks.

**Evaluation** We evaluate the in-context learning abilities of the AutoCompressor based on Llama-2-7B from Section 4.3 on eleven classification and multiple-choice question-answering datasets. For each dataset, we evaluate the effect of compressing 1, 2 or 3 segments of demonstrations into 50, 100 or 150 summary vectors. For each segment, we include as many demonstrations as possible until we reach 750 tokens. For SST-2, this corresponds to 30 demonstrations per segment on average. We compare this compression approach with the results obtained by prompting the model using 150 and 750 tokens' worth of plain-text demonstrations.

We use contextual calibration (Zhao et al., 2021) and class-balanced sampling when these techniques improve performance on a validation set. For each dataset, we report the mean accuracy and standard deviation over 7 random seeds. The detailed settings for each dataset can be found in Table 11. In Table 12 in the Appendix, we also compare the ICL performance of our OPT-2.7B based AutoCompressor models against the RMT baseline and a pre-trained OPT-2.7B, and include the performance of the pre-trained Llama-2-7B model.

**Results** We show evaluation results in Table 4. Results show that summary vectors consistently improve performance over the zero-shot baseline. Furthermore, summary vectors increase accuracy

| | AG News | SST-2 | BoolQ | WIC | WSC | RTE | CB | COPA | MultiRC | MR | Subj |
|---|---|---|---|---|---|---|---|---|---|---|---|
| Zero-shot | $63.3_{(0.0)}$ | $67.7_{(0.0)}$ | $67.4_{(0.0)}$ | $50.8_{(0.0)}$ | $43.3_{(0.0)}$ | $58.8_{(0.0)}$ | $42.9_{(0.0)}$ | $52.5_{(0.0)}$ | $52.5_{(0.0)}$ | $57.4_{(0.0)}$ | $49.3_{(0.0)}$ |
| 50 summary vecs. | $79.6_{(4.9)}$ | $\mathbf{94.2}_{(1.6)}$ | $\mathbf{70.1}_{(3.3)}$ | $51.6_{(2.1)}$ | $47.7_{(8.7)}$ | $66.3_{(7.0)}$ | $46.4_{(18.7)}$ | $84.5_{(1.0)}$ | $52.6_{(2.8)}$ | $91.5_{(1.0)}$ | $53.5_{(3.6)}$ |
| 100 summary vecs. | $\mathbf{87.6}_{(1.2)}$ | $92.6_{(3.3)}$ | $66.3_{(2.8)}$ | $52.5_{(2.2)}$ | $42.9_{(2.5)}$ | $63.5_{(6.6)}$ | $\mathbf{64.5}_{(5.9)}$ | $85.9_{(0.4)}$ | $\mathbf{56.1}_{(1.2)}$ | $90.7_{(2.6)}$ | $57.0_{(5.6)}$ |
| 150 summary vecs. | $85.4_{(3.4)}$ | $92.3_{(2.9)}$ | $68.0_{(1.8)}$ | $\mathbf{52.8}_{(1.5)}$ | $49.9_{(7.6)}$ | $65.3_{(6.6)}$ | $54.8_{(5.8)}$ | $\mathbf{86.1}_{(0.6)}$ | $54.8_{(2.2)}$ | $91.1_{(2.2)}$ | $56.6_{(7.9)}$ |
| ICL (150 tokens) | $74.5_{(2.2)}$ | $92.4_{(3.1)}$ | $67.4_{(0.0)}$ | $52.4_{(2.7)}$ | $\mathbf{51.8}_{(6.9)}$ | $69.1_{(2.1)}$ | $46.4_{(23.0)}$ | $80.0_{(1.9)}$ | $52.5_{(0.0)}$ | $79.7_{(15.7)}$ | $57.9_{(10.7)}$ |
| ICL (750 tokens) | $81.2_{(4.1)}$ | $93.8_{(1.2)}$ | $67.7_{(2.7)}$ | $52.4_{(2.0)}$ | $40.0_{(5.7)}$ | $\mathbf{73.1}_{(3.5)}$ | $50.3_{(2.8)}$ | $82.6_{(1.6)}$ | $47.0_{(3.2)}$ | $\mathbf{91.6}_{(0.8)}$ | $\mathbf{60.7}_{(14.8)}$ |

Table 4: Evaluation of the ICL performance of the Llama-2 7B model. Each summary is 50 tokens-long and corresponds to a segment of 750 tokens' worth of demonstrations. We also report accuracies when prompting the AutoCompressor with 150 and 750 tokens' worth of plaintext demonstrations as baselines. Note that for BoolQ and MultiRC, demonstrations are too long to fit into 150 tokens.

compared to 150 tokens worth of plain demonstrations on 8/11 tasks. On 8 tasks (AG News, SST-2, BoolQ, WiC, WSC, CB, COPA and MultiRC), summary vectors also out-perform ICL with 750 tokens' worth of plain text demonstrations. Summary vectors emerge as a strong alternative to plain text demonstrations, as they increase accuracy while reducing inference cost.

In Table 12 (Appendix E), we find that the OPT-2.7B AutoCompressor achieves higher accuracies than the RMT baseline on 8 out of 11 tasks and that the RMT model does not benefit from multiple compression steps. This shows that summary accumulation is an effective mechanism for compressing in-context demonstrations. We also observe that our fine-tuned Llama-2 AutoCompressor has substantially worse zero-shot accuracy on some tasks compared to the Llama-2 initialization, and slightly worse ICL performance. We suspect that this is due to domain mismatch in our fine-tuning data and the Llama-2 pre-training corpus.

# 6 Compressing Retrieval Corpora for Efficient Inference

We study the usefulness of pre-computing summary vectors for large collections of documents. These can be stored and later retrieved for efficient inference. Since inference is typically more expensive than storage, this approach has the potential to achieve good practical trade-offs.

## 6.1 Retrieval-augmented Language Modeling

Retrieval-augmented language models improve token predictions by retrieving information from a data store. A number of approaches have been proposed to infuse external knowledge in the input layer (Guu et al., 2020; Shi et al., 2023), intermediate layers (Borgeaud et al., 2022) or at the output layer (Khandelwal et al., 2020; Zhong et al., 2022).

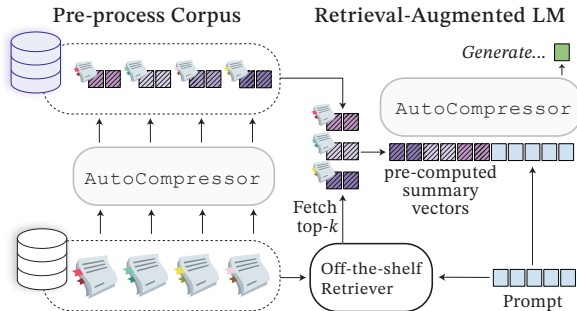

Figure 3: Efficient retrieval-augmented language modeling with AutoCompressors. Large corpora can be pre-processed into compressed summary vectors which can be stored cheaply. Upon retrieval, compressed summaries are fused for efficient access to multiple documents in a single forward pass.

**REPLUG** Our case study focuses on REPLUG (Shi et al., 2023), which is a simple method for combining a pre-trained language model with an off-the-shelf retriever to improve language modeling performance. Given access to an external corpus $\mathcal{C}$, REPLUG retrieves $k$ passages $\mathcal{D} = \{d_1, \ldots, d_k\}$ based on a segment $x$ to score the next segment $y$. The overall probability for $y$ is computed by ensembling the predictions based on different passages:

$$p(y \mid x, \mathcal{D}) = \sum_{d \in \mathcal{D}} \lambda(d, x) \cdot p(y \mid \text{CONCAT}(d, x)),$$

where $\lambda(d, x)$ are the normalized similarity scores from the retriever and $\text{CONCAT}(d, x)$ denotes concatenation of $p$ and $x$. This method incurs a substantial overhead, since it requires $k$ forward passes over sequences $\text{CONCAT}(d, x, y)$.

**Fused Summaries** We introduce a setting for retrieval-augmented language modeling close to fusion-in-decoder (Izacard and Grave, 2021). We concatenate the summary vectors of retrieved passages $\mathcal{D}$ to form the *fused summary vectors*, $\sigma_{\mathcal{D}} = \text{CONCAT}(\sigma_{d_k}, \ldots, \sigma_{d_1})$, where $d_k, \ldots, d_1$ are ordered from least-to-most relevant. This resembles

| Passages | | Perplexity Gain (%) | | | | Throughput (examples/s) | | | |
|---|---|---|---|---|---|---|---|---|---|
| | | top-1 | top-2 | top-5 | top-10 | top-1 | top-2 | top-5 | top-10 |
| 50 tokens | REPLUG | -0.64 | 0.58 | 1.68 | 2.35 | 51 | 38 | 16 | 9 |
| 50 tokens | Fused Passages | 0.71 | 1.01 | 1.70 | 2.60 | 28 | 27 | 23 | 17 |
| 512 tokens → 50 sum. vecs. | Fused Summaries | **1.04** | **1.67** | **2.63** | **3.74** | 28 | 27 | 23 | 17 |
| 512 tokens | REPLUG | -1.47 | 2.24 | 5.25 | 8.30 | 18 | 10 | 6 | 3 |

Table 5: PPL gains (%) from different retrieval-augmented language modeling settings, over the no-retrieval baseline. We evaluate the OPT-2.7B AutoCompressor and we report throughput on a single NVIDIA A100 GPU for each method without batching examples. Fused Summaries outperforms Fused Passages and REPLUG with 50-token passages. Moreover, Fused Summaries top-10 outperforms REPLUG top-2 with 512-token passages while also gaining a $1.7\times$ throughput increase.

summary accumulation as described in Section 3. We also find it useful to smooth probability scores and re-order the retrieved passages based on their summary vectors (Appendix F). Figure 3 gives an overview of our approach.

**Fused Passages** We establish a baseline for fusing summary vectors by concatenating the plain-text passages and computing smoothed probabilities, see Appendix F. Unlike summary vectors, this method is limited by the model's context window.

**Experiments** We evaluate the OPT-2.7B Auto-Compressor introduced in Section 4.1 without any additional fine-tuning. Similar to Shi et al. (2023), we retrieve from the Pile. We use Books3, FreeLaw, GitHub, Wikipedia, Gutenberg, ArXiv, HackerNews, and YoutubeSubtitles. We index 10B tokens for each domain, which are split into passages of 512 or 50 tokens.

We sample segments of 256 tokens from the Pile validation data, using the first 128 tokens as context $x$ for retrieval and the last 128 tokens $y$ for evaluation. We use the Contriever model (Izacard et al., 2022) for retrieval, and retrieve the top 10 passages. We also deduplicate our data by removing passages that overlap with $x$ by 64 tokens.

**Results** Results are shown in Table 5. We find that Fused Summaries outperforms Fused Passages and REPLUG when 50-token passages are retrieved. We measure throughput empirically and show that for 10 retrieved documents, Fused Summary Vectors remains inexpensive. We note that compressing the 10B token datasets results in disk space of 5TB per domain when stored in half-precision format.[5] Therefore Fused Summaries achieves a good trade-off between storage costs and throughput.

Moreover, Fused Summaries outperforms RE-PLUG top-2 with 512-token passages and sees a 1.7x throughput increase, which shows that the model benefits from the diversity of compressed documents. However, REPLUG top-10 outperforms Fused Summaries. We leave it as future work to explore how to produce higher quality summary vectors to better utilize the compressed passages.

We note that fusing summary vectors is effective despite a mismatch in training since we draw independent summary vectors from separate documents. Furthermore, our AutoCompressor model is only ever trained to accumulate 3 sets of summary vectors, and yet it benefits from fusing the summary vectors of up to 10 documents.

### 6.2 Unsupervised Passage Re-ranking

Finally, we consider the case study of passage re-ranking, in which a fast off-the-shelf retriever like BM25 retrieves a large set of candidate passages, and a more capable re-ranker refines the ranking to increase the rank of the most relevant passages.

**Method** Sachan et al. (2022) introduce an effective method for leveraging language models as re-rankers with no additional supervision or fine-tuning. Given a query $q$ and a set of candidate passages $\{p_1, \ldots, p_k\}$, the language model scores the likelihood of the query $q$ conditioned on the prompt "`Passage: {`$p_i$`}. Please write a question based on this passage.`" for each passage $p_i$ and re-ranks the passages based on the scores.

**Experiments** We consider the task of re-ranking BM25 passages on the NQ test set (Balachandran et al., 2021) and compare out-of-the-box AutoCompressors with 20 and 50 summary tokens to pre-trained OPT models from 125M to 2.7B parameters. We pre-compute summary vectors for 21M passages from a Wikipedia corpus (Karpukhin et al.,

---

[5] For comparison, storing the transformer output at every single token (e.g., in an encoder-decoder setting) would take up 51 TB, and storing all attention states would be 3,276 TB.

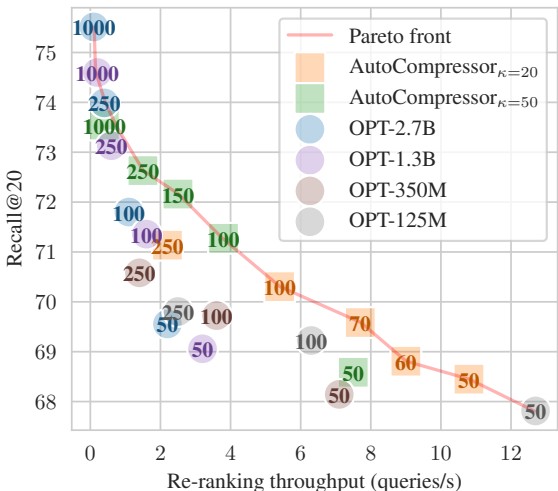

Figure 4: We compare AutoCompressors (squares) in an unsupervised passage re-ranking setting to pre-trained language models (circles). The number on each data point shows how many passages retrieved by BM25 are re-ranked, and the vertical axis shows the Recall@20 performance of the re-ranking system on the NQ test set. We consider the throughput on a single NVIDIA A100 GPU and assume that multiple queries cannot be batched. By leveraging pre-computed summary vectors for passages, AutoCompressors lead to re-ranking solutions that lie on the Pareto front of recall vs. compute.

2020), which requires 2.1TB and 5.4TB disk space in half precision for 20 and 50 summary vectors respectively. We measure the quality of the re-ranked results using Recall@20.

**Results** The results are shown in Figure 4. We measure throughput for individual un-batched queries on a single NVIDIA A100 80GB GPU and assume that the latency of loading summary vectors is negligible. Although the passages are only 100 words long, resulting in low compression rates, summary vectors substantially speed up the inference, while sacrificing on performance less than smaller models. This leads to a Pareto-optimal trade-off between compute and performance and demonstrates that summary vectors often retain sufficient information from a passage to assess its relevance for a particular query.

## 7   Conclusion

We have introduced a training strategy for adapting pre-trained LMs into AutoCompressors, which recursively compress contexts into summary vectors. Our experiments indicate that summary vectors retain important contextual information, that they can encode in-context demonstrations, and that they

can be used in retrieval settings. Summary vectors can also be pre-computed, cached and re-used. This offers practical efficiency gains by reducing the size of the attention window. Significant future work remains in scaling AutoCompressors to bigger models and improving the quality of summary vectors to further close the gap with full attention over long-range contexts.

## Limitations

1. We only apply AutoCompressors to OPT models of up to 2.7B parameters and a Llama model of 7B parameters. Future work needs to establish how AutoCompressors perform for even larger models. As the summary vector dimension grows, there is promise for retaining more information per vector.
2. Our results suggest that summary vectors ignore some useful information that is accessible via full attention. Additionally, models do not always benefit from increasing the number of summary vectors. We suspect that the training signal for learning summary vectors efficiently might be limited by pre-trained models being very good at making predictions from the plaintext tokens in the current segment. Future work is needed to improve this optimization.
3. Summary accumulation still leads to quadratic complexity with increasing number of segments, albeit at a much lower rate than full attention. Future work may explore ways to combine many summary vectors more efficiently.

## Acknowledgments

We thank Mengzhou Xia, Howard Chen, Vishvak Murahari, Aatmik Gupta, Zirui Wang, Jiatong Yu, and the members of the Princeton NLP group for helpful discussion and valuable feedback. This research is supported by an NSF CAREER award (IIS-2239290), a Sloan Research Fellowship, and a Data Science Research Award from Adobe. AC also gratefully acknowledges support from the Minerva Research Foundation.

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

# A   Models and Data

All models are fine-tuned from OPT models on the Pile. We conduct our experiments using a single NVIDIA A100 80GB GPU and we use Flash Attention (Dao et al., 2022) as an efficient implementation of exact attention over long sequences. We also use gradient checkpointing between compressed segments to reduce GPU memory.

## A.1   OPT Experiments on 8K Tokens

We fine-tune our models on 2B tokens from the Pile. We sample 500M tokens from the following Pile subdomains: Books3, FreeLaw, GitHub and Wikipedia.

The following models use a learning rate of 2e-5, a batch size of 130K tokens, 1,00 warm-up steps, and the Adam optimizer (Kingma and Ba, 2015):

1. The fine-tuned OPT-2.7B baseline is fine-tuned on documents of up to 2,048 tokens.
2. The extended full-attention baseline is fine-tuned on documents of up to 4,096 tokens by extending the positional embeddings of OPT-2.7B to 4,096 positions. We initialize the embeddings for positions [2049..4096] with the embeddings for positions [1..2048].
3. The RMT baseline is fine-tuned on documents of up to 8,192 tokens. Each document is segmented into four segments of 2,048 tokens. We use $\kappa = 50$ summary vectors but we do not use summary accumulation, randomized segmenting, or stop-gradients.
4. Our AutoCompressor is fine-tuned on documents of up to 6,144 tokens. Each document is randomly segmented into four segments such that the first two segments add up to 3,072 tokens. The length of each segments ranges from 1,024 to 2,048 tokens. We use $\kappa = 50$ summary vectors and summary accumulation. We stop gradients every two compression steps.

All models are evaluated on documents sampled from the Pile with a fixed length of 8,192 tokens. We sample 610 documents from each of the following domains: Books3, FreeLaw, GitHub, Wikipedia (in-domain), and ArXiv, Gutenberg, HackerNews, YoutubeSubtitles (out-of-domain). Examples of documents from each of those domains can be found in Tables 9 and 10.

## A.2   OPT Experiments on 30K Tokens

We fine-tune our models on 2 billion tokens from the Books3 subdomain of the Pile. All models are fine-tuned on documents of up to 30,720 tokens. We use a learning rate of 2e-5, a batch size of 130k tokens, 1,000 warm-up steps and the Adam optimizer.

1. RMT-1.3B uses $\kappa = 50$ summary vectors and is fine-tuned without summary accumulation, randomized segmenting, or stop-gradients. Each document is split into 15 segments of 2,048 tokens Even with gradient checkpointing, attempting to fine-tune a 2.7B parameter RMT model on this dataset leads to an out-of-memory error.
2. The AutoCompressor models are fine-tuned from OPT-1.3B and 2.7B on documents of up to 30,720 tokens. Each document is split into 20 segments such that segment $2i$ and segment $2i+1$ add up to 3,072 tokens. The length of each segment is randomly sampled between 1,024 and 2,048. We use $\kappa = 50$ summary vectors with summary accumulation and we stop gradients every two compression steps.

All models are evaluated on documents of 30,720 tokens from the Pile. We use 1,000 documents from Books3 (in-domain) and 1,000 documents from Gutenberg (out-of-domain).

## A.3   Llama-2 Experiments on 8K Tokens

We fine-tune our Llama-2 models on 15B tokens from RedPajama. We sample 1B tokens from long documents in ArXiv, Books, C4, GitHub, as well as 10B tokens from CommonCrawl, 800M from Wikipedia and 70M tokens from StackExchange.

Both our AutoCompressor and our Extended Full Attention baseline are fine-tuned from Llama-2-7B on sequences of 6,144 tokens with LoRA (Hu et al., 2022) parameter efficient fine-tuning applied to the attention heads. We use a LoRA dimension of 16 applied to the QKV- and Out-projections. We use a learning rate of 4e-4, a batch size of 200K tokens, 5,000 warm-up steps and the Adam optimizer. For the AutoCompressor, we also optimize the newly initialized summary token embeddings.

We train our AutoCompressor in the same way as the OPT-2.7B AutoCompressor, with $\kappa = 50$, randomly segmenting each sequence into four sengents, and stopping gradients every two compression steps. The Extended Full Attention baseline is fine-tuned with a RoPE $\theta$ value of 80,000.

We evaluate our models on 500 sequences of 8,192 tokens from each of ArXiv, Books, C4, GitHub, StackExchange, and 5,000 sequences from CommonCrawl.

## B    No-context Language Modeling

In Table 6, we verify that our fine-tuning strategy does not significantly affect the language modeling capabilities of the OPT AutoCompressors when no summary tokens are given. We find that the Auto-Compressor performs slightly better than the RMT model and significantly better than the extended full attention model when no additional context is given. Moreover, the AutoCompressor almost matches the OPT02.7B fine-tuned baseline, with perplexity increasing by less than 1%.

|  | In-domain | Out-of-domain |
|---|---|---|
| OPT-2.7B | 7.53 ↑19.9% | 9.19 ↑7.7% |
| OPT-2.7B fine-tuned | 6.28 | 8.53 |
| AutoCompressor-2.7B | 6.31 ↑0.5% | 8.60 ↑0.8% |
| RMT-2.7B | 6.34 ↑1.0% | 8.62 ↑1.1% |
| Extended full attention | 6.57 ↑6.4% | 8.94 ↑4.8% |

Table 6: Held-out perplexity of all models on 2048 tokens without summary vectors or additional context.

## C    AutoCompressor Ablations

We train OPT AutoCompressor models as in Section 4.1 while varying $\kappa = 20, 50, 70, 100$. In Table 7, we report the perplexity evaluation on documents of 8192 tokens across all evaluation domains.

| | Compressed tokens | | | |
|---|---|---|---|---|
| $\kappa$ | 0 | 2048 | 4096 | 6144 |
| 20 | **7.36** | 7.05 | 7.01 | 7.00 |
| 50 | 7.37 | **6.99** | **6.94** | **6.93** |
| 70 | 7.41 | 7.01 | 6.97 | 6.95 |
| 100 | 7.48 | 7.07 | 7.01 | 7.00 |

Table 7: Held-out perplexity across all evaluation domains for AutoCompressors based on OPT-2.7B trained with different numbers of summary tokens $\kappa$. We observe that $\kappa = 50$ performs the best overall.

## D    Token-level AutoCompressor Analysis

In Figure 5, we plot the perplexity gains achieved by the OPT AutoCompressor and the extended full attention baseline from Section 4.1 over the pre-trained OPT-2.7B model. We plot the gains achieved by the AutoCompressor both without any additional context and with the summary vectors obtained from 2048 compressed tokens.

Results show that the summary vectors help reduce perplexity over the entire 2,048-token segment. This shows that summary vectors do not only contain information which helps continue the previous sequence.

Figure 5 also shows that the extended full-attention baseline benefits more from the additional 2,048 context tokens than the AutoCompressor at the start of the sequence, but that the AutoCompressor achieves stronger gains at the end of the sequence. This shows that summary vectors effectively capture long-range textual dependencies and that fine-tuning AutoCompressors produces more robust models than fine-tuning extended full-attention models.

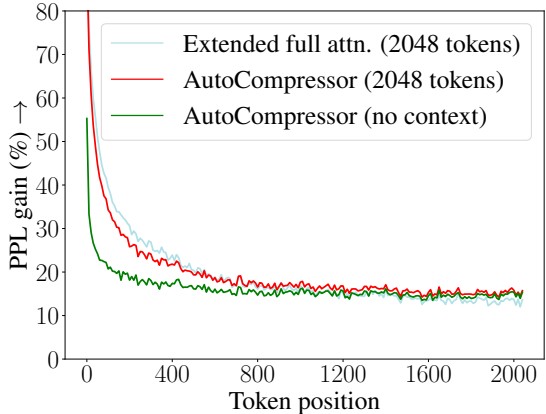

Figure 5: We plot the perplexity gain over OPT-2.7B for our AutoCompressor model and the 4096-extended attention baseline. We track the perplexity at each token position in sequences of 2048 tokens. The Auto-Compressor model almost matches the strong extended-attention baseline at the start of sequences and outperforms it at the end of sequences.

In Tables 9 and 10, we give hand-picked examples of sequences from each evaluation domain, highlighting which tokens benefit the most from the compressed context. We compress the first 300 tokens in every document from the evaluation set and evaluate on the following 100 tokens. In the notation of Section 3.1, we measure the perplexity gain of each token as

$$\frac{p(x_t^2 \mid x_1^2, \ldots, x_{t-1}^2, \sigma_1)}{p(x_t^2 \mid x_1^2, \ldots, x_{t-1}^2)}.$$

For each example, we record the top 3-5 most improved token predictions.

We find that the tokens which benefit the most from the summary vectors are often interpretable. Names of characters, dates, and locations are often

copied through the summary vectors (see the examples for Wikipedia, FreeLaw, or HackerNews). We also find that the model is able to reason over the summary vectors, as the tokens which benefit the most are sometimes not explicitly present in the compressed context, but are closely associated with the domain of speech (see the examples for Books3, Gutenberg and YoutubeSubtitles.). Finally, we find that summary vectors are often useful for continuing the previous sentence (see the GitHub example.)

# E   In-Context Learning Details

We evaluate on in-context examples of the following datasets: AG News (topic classification, Zhang et al. (2015)), SST-2 (sentiment analysis, Socher et al. (2013)), BoolQ (Boolean Questions, Clark et al. (2019)), WiC (Word-in-Context, word sense dismabiguation, Pilehvar and Camacho-Collados (2019)), WSC (Winograd Schema Challenge, coreference resolution, Levesque et al. (2012)), RTE (Recognizing Textual Entailment, Dagan et al. (2005); Haim et al. (2006); Bentivogli et al. (2009)), CB (CommitmentBank, de Marneffe et al. (2019)), COPA (Choice of Plausible Alternatives, Roemmele et al. (2011)), MultiRC (Multi-Sentence Reading Comprehension, Khashabi et al. (2018)), MR (Movie Reviews, Pang and Lee (2005)), Subj (Subjectivity, Pang and Lee (2004). We follow the GPT-3 prompt templates (Brown et al., 2020) and detail our evaluation setting for OPT and Llama-2 in Table 11.

In Table 12, we compile evaluation results for OPT-2.7B, Llama-2-7B, as well as our AutoCompressor and RMT models.

# F   Fused Retrieval-augmented Language Modeling

| | Perplexity Gain (%) | | | |
|---|---|---|---|---|
| Passages | top-1 | top-2 | top-5 | top-10 |
| Fused Summaries | 1.04 | **1.67** | **2.63** | **3.74** |
| Fused Summaries w/o re-ranking | 1.04 | 1.52 | 2.02 | 2.63 |

Table 8: PPL gains (%) over the no-retrieval baseline for Fused Summary with and without re-ranking. In re-ranking, we order the passages based on the $\ell_2$ norms of their summary vectors before concatenating the summary vectors, whereas w/o re-ranking we use the retrieval scores from the Contriever model. Re-ranking consistently produces higher perplexities.

We provide details and ablations for our proposed REPLUG alternative. Inspired by fusion-in-decoder (Izacard and Grave, 2021), we fuse summary vectors or passages in a single forward pass.

**Fused Summary Vectors**  The summary vectors of retrieved passages $\mathcal{D}$ are concatenated in order of increasing retrieval scores to form *fused summary vectors*, $\sigma_{\mathcal{D}} = \text{Concat}[\sigma_{d_k}, \ldots, \sigma_{d_1}]$. This resembles summary accumulation as described in Section 3, but differs in that the retrieved summary vectors were produced independently rather than recursively. Nevertheless, we find that AutoCompressors transfer well to this setting.

Furthermore, we find it beneficial to smooth the conditioned probabilities with the unconditioned probabilities $p(y \mid x)$, and compute

$$p(y \mid x, \mathcal{D}) = \frac{p(y \mid \text{Concat}[\sigma_{\mathcal{D}}, x]) + p(y \mid x)}{2}.$$

We also show that language-modeling performance improves when $\mathcal{D}$ is re-ordered based on the smallest $\ell_2$ distance between the summary vectors $\{\sigma(d_1), \ldots, \sigma(d_k)\}$ and $\sigma_x$. This incurs negligible overhead since $\sigma_x$ can be constructed during the same forward pass which computes $p(y \mid x)$. The ablation for this is shown in Table 8

**Fused Passages**  We establish a baseline for Fusing Summary Vectors by concatenating the corresponding plain-text passages $D = \text{Concat}[d_k, \ldots, d_1]$ and computing

$$p(y \mid x, \mathcal{D}) = \frac{p(y \mid \text{Concat}[D, x]) + p(y \mid x)}{2}.$$

Note that this approach is quickly limited by the size of the pre-trained language model's context window, especially when retrieving many long passages.

| Domain | Compressed context | Evaluation sequence | Most improved tokens |
|---|---|---|---|
| Books3 | Surrealism—not for Breton's depreciation of "Red Front," but for a seemingly insignificant aside. In early March, before sending the text to press, Breton showed it to Aragon. The latter consented to the publication, with one exception: a footnote in which Breton quoted the PCF official's remark (which Aragon had earlier reported to him) about "complicating the simple, healthy relations between men and women"—a clear illustration, Breton felt, of "just how much bad faith or mental indigence we were up against." Aragon considered internal Party statements to be confidential, and asked that the footnote be removed; according to him, Breton "spontaneously crossed out the note on the galleys with a delete mark that I can still recall... saying that he wanted to give the Party no excuse for expelling me." But when _The Poverty of Poetry_ came off press the next day, the incriminating footnote was still there.

Whether Breton retained the note as a test of Aragon's loyalty, or whether he deemed this example of PCF stupidity too good to waste, or whether the printer simply neglected to make the correction, no one has ever established. But the result was that this single act came to represent for Aragon every philosophical difference, stricture, and humiliation that had ever darkened his long friendship with Breton. On March 10, he responded to the tract via an anonymous note in | _L'Humanité_ : "Our comrade Aragon informs us that he has absolutely nothing to do with the publication of a pamphlet entitled _The Poverty of Poetry_... He wishes to make it clear that he entirely disavows both the contents of this pamphlet and the attention it has drawn to his name, every Communist being duty-bound to condemn the attacks contained in this pamphlet as incompatible with the class struggle." This short paragraph was the only notice he ever saw fit to give of | Poverty
Po
Ar
agon
Human |
| Wikipedia | Shi Ce
Shi Ce (; born 15 December 1985) is a Chinese deaf female table tennis player. She has represented China at the Deaflympics four times from 2005-2017. Shi Ce has been regarded as one of the finest athletes to have represented China at the Deaflympics, having won 14 medals at the event since making her debut in the 2005 Summer Deaflympics.
Biography
Shi Ce was born in Yichun, Heilongjiang on 15 December 1985. She was born with an ear condition that impaired her hearing which resulted in her deafness and has congenital malformation in her right ear. Her parents decided to consult a doctor and took her to an hospital in the Zhejiang Province in order to cure her ear impairment when she was just five years old. The doctor suggested that surgery would cause facial paralysis after Shi Ce's parents demanded for a surgery. Shi Ce took the sport of Table tennis and started playing it at the age of nine.
Career
Shi Ce has won 14 medals in her Deaflympic career as a Table tennis player including 11 gold medals. Shi Ce was eligible to compete at the National Games of China despite her deafness, in 2015. In the competition, she secured gold medals in singles, doubles, mixed doubles and in the team events.
2005 Summer Deaflympics Shi Ce made her first appearance at an international sports | event during the 2005 Summer Deaflympics and excelled on her debut Deaflympic event after winning gold medals in the women's singles, doubles and in the mixed doubles. She was also the part of the Chinese Table tennis team which secured the silver medal in the 2005 Deaflympics. In the same year, she received the Deaf Sportswoman of the Year award from the ICSD for her remarkable performances at the 2005 Summer Deaflympics. Shi Ce | Ce
De
2005
Summer
Shi |
| Github |  import sys
import datetime
def basic(arguments):
    import api
    critic = api.critic.startSession(for_testing=True)
    repository = api.repository.fetch(critic, name="critic")
    branch = api.branch.fetch(critic,
        repository=repository, name=arguments.review)
    review = api.review.fetch(critic, branch=branch)
    alice = api.user.fetch(critic, name="alice")
    bob = api.user.fetch(critic, name="bob")
    dave = api.user.fetch(critic, name="dave")
    erin = api.user.fetch(critic, name="erin")
    all_comments = api.comment.fetchAll(critic)
    assert isinstance(all_comments, list)
    EXPECTED = { 0: { "text": "This is a general issue.", "location": None, | "type": "issue", "state": "open" },
1 : {"text": "This is a general note.",
    "location": None,
    "type": "issue", | 1
location
}
text
issue |
| FreeLaw | 8
By the end of 1975, Farmers National was insolvent and under investigation by the Florida Department of Insurance. The Miami newspapers published a series of articles describing the relationship between Hauser and the company. Lawrence Lee, an attorney for an Arizona union group, investigated Farmers National in connection with an Old Security-Farmers National proposal. He was told by the Florida insurance department that Farmers National was connected with Hauser, that it had been injected with questionable assets which were being investigated by the department, and that it had been fined $5,000 for failing to disclose both Hauser's ownership and a loan to one of its directors. Lee contacted Richard Halford, vice-president at Old Security in charge of union group insurance, and related the information he had received. Halford assured Lee that he was aware of Hauser's reputation, but that Hauser was no longer involved with Farmers National. Halford then called Kavanagh, who told him that Hauser had no official capacity with the company, and that the financial problems had been cleared up. Halford did not attempt to check the accuracy of Kavanagh's representations with the Florida Department of Insurance.
9
Hauser controlled a second company, Family Provider Life Insurance Company ("Family Provider"). In 1975, the company had no business, no office, and assets of $50,000. Because of Farmers National's insolvency, Hauser decided to activate | Family Provider, and its assets were increased to $250,000, the minimum required to conduct business in Arizona, where the company was licensed. In January 1976, Boden and Kavanagh met with Halford and Robert Barton, president of Old Security, to propose a new agreement between Old Security and Family Provider for the purpose of obtaining the Fund business. Both Barton and Halford considered Family Provider and Farmers National to be "synonymous" and believed that Kavanagh and Boden | Security
Old
Family
avan
assets |

Table 9: Examples of sequences from in-domain test Pile domains. We highlight the tokens from the evaluation sequence which benefit the most from the summary vectors. In Books3, *L'Humanité* is prominent French newspaper associated with Breton and his circle. In GitHub, the summary vectors carry information about the logical and syntactical continuation of the context.

| Domain | Compressed context | Evaluation sequence | Most improved tokens |
|---|---|---|---|
| HackerNews | Hackers steer Tesla into oncoming traffic by placing three stickers on the road - velmu https://www.businessinsider.com/tesla-hackers-steer-into-oncoming-traffic-with-stickers-on-the-road-2019-4 
 ====== chrisbolt From yesterday: 
 [https://news.ycombinator.com/item?id=19536375] 
 (https://news.ycombinator.com/item?id=19536375) 
 —— 
 gregmac 
 While I'm hugely skeptical of the current state of self-driving cars, you could probably get human drivers to make the same mistake if you were to repaint the lines. However, humans will also notice the oncoming cars (if there are any) and avoid getting in a head-on collision. The thing missing from this test is that critical practical piece: if there was an oncoming car, will the Tesla do something to avoid the collision? I would assume that not getting in a head-on crash is higher priority than staying in the lane markings. 
 Without oncoming traffic, all this is testing is what the Tesla considers valid line markings. I'm sure there's room for improvement here (such as checking where the other lane is, raising the requirement for how well-defined the lines have to be, etc), but | those are also going to involve trade-offs where there are legitimate situations that will stop working. 
 I think you could just as easily title this video "Tesla auto-pilot follows road markings even if they're really bad". 
 Edit: The best shot I could get from the video [1] makes me even more upset at this test: these look like the temporary markings often used during construction, just before they come and paint the normal lines using the big | Tesla 
 test 
 markings 
 auto |
| ArXiv | $z_k = h_k(x_k) + v_k, \qquad v_k \sim \mathcal{N}(0, R_k)$ 
 In the above equations, we see that the transition matrix $F_{k,k-1}$ has been replaced by the nonlinear vector-valued function $f_{k,k-1}(\cdot)$, and similarly, the matrix $H_k$, which transforms a vector from the state space into the measurement space, has been replaced by the nonlinear vector-valued function $h_k(\cdot)$. The method proposed by the Extended Kalman Filter is to linearize the nonlinearities about the current state prediction (or estimate). That is, we choose $F_{k,k-1}$ as the Jacobian of $f_{k,k-1}$ evaluated at $\hat{x}_{k-1|k-1}$, and $H_k$ as the Jacobian of $h_k$ evaluated at $\hat{x}_{k|k-1}$ and proceed as in the linear Kalman Filter of Section $sec :: kf$.[18] Numerical accuracy of these methods tends to depend heavily on the nonlinear functions. If we have linear constraints but | a nonlinear $f_{k,k-1}(\cdot)$ and $h_k(\cdot)$, we can adapt the Extended Kalman Filter to fit into the framework of the methods described thus far. Nonlinear Equality and Inequality Constraints 
 —— 
 Since equality and inequality constraints we model are often times nonlinear, it is important to make the extension to nonlinear equality and inequality constrained Kalman Fil | Extended 
 linear 
 h 
 k 
 Kal |
| Gutenberg | eight or nine cents. Telegrams in foreign languages are sent within the empire for five sen per word, with a minimum charge of twenty-five sen for five words or a fraction thereof. No charge is made for delivery within a radius of 2-1/2 miles of the telegraph office. 
 There are no private telegraph corporations. The government builds, owns, and operates the lines just as it does the mails. The postal and 101 telegraph systems are intimately connected, and the same office does service for both. 
 The first telegraph line in Japan was opened in 1869. The venture proving a success, the following year the line was extended and a general telegraphic system for the whole country decided upon. The rapid construction of telegraph lines began in 1872, from which year it has gone forward uninterruptedly. At present the lines extend to every corner of the empire. The first lines were surveyed, built, and operated under foreign experts; but the natives have learned so rapidly that they have been enabled to do away with all foreign employees. All of the materials and instruments in use, with the exception of submarine cables and the most delicate electrical measuring apparatus, are made in Japan. 
 MAILS.–The Japanese mail system was modeled after the American in 1871. | At first it was limited to postal service between the three large cities of Tokyo, Kyoto, and Osaka; but in 1872 it was extended to the whole country, with the exception of a certain part of the Hokkaido, which was without roads and almost without population. To-day there is no village or hamlet in the whole land which does not enjoy the convenience of a good postal system. The mails are sent with promptness and | limited 
 postal 
 Tokyo |
| YoutubeSubtitles | te que no voy a esa escuela." 
 Johnny Galecki 
 El Dr. Leonard Hofstadter obtuvo su doctorado a los 24 años, pero el actor que lo interpreta sólo llegó a medio camino de la secundaria. En una entrevista con Time Out Chicago en el 2009, Johnny Galecki reveló que abandonó la escuela a mediados del octavo grado luego de años de evitar ir a clases a toda costa. Le dijo a Time Out, "Una vez que las divisiones largas aparecieron en tercer grado, iba al baño por 45 minutos y nadie lo notaba, todos los días a la misma hora del día, sólo para escapar de ellas." Puede que Galecki no tenga un cerebro matemático, pero siempre tuvo inteligencia callejera. "El conocimiento es el mejor y más seguro tesoro... Vaya, me aburro a mí mismo." A los 14 años, vivió solo en un apartamentito en Burbank, California, mient | ras trabajaba en la comedia American Dreamer, su primer gran trabajo. Su familia pasó nueve meses en Long Beach antes de regresar a Chicago, y él se quedó para concentrarse en su carrera como actor. 
 Jim Parsons 
 El Dr. Sheldon Cooper fue un niño prodigio. Comenzó la universidad cuando tenía 11 años | Parsons 
 aba 
 Jim 
 Dr |

Table 10: Examples of sequences from out-of-domain test Pile domains. We highlight the tokens from the evaluation sequence which benefit the most from the summary vectors. In Gutenberg, 'Tokyo' is not copied over from the compressed context but is inferred from the discussion of Japan. In YoutubeSubtitles, 'Jim Parsons' benefits the most from the summary vectors because the context discusses his co-star John Galecki in *The Big Bang Theory*.

| Dataset | Prompt template | OPT-based models | | | Llama-2-based models | | |
|---|---|---|---|---|---|---|---|
| | | Toks. / dem. | Cal. | Bal. | Toks. / dem. | Cal. | Bal. |
| AG News | `Article: {text}\nTopic: {label}` | 65 | ✓ | | 75 | ✓ | |
| SST-2 | `Sentence: {sentence}\nSentiment: {label}` | 22 | ✓ | ✓ | 25 | ✓ | ✓ |
| BoolQ | `{passage}\nquestion: {question}?\nanswer: {label}` | 165 | ✓ | | 170 | ✓ | |
| WiC | `{sentence1}\n{sentence2}\nquestion: Is the word '{word}'` `used the same way in the two sentences above?\nanswer: {label}` | 45 | ✓ | | 45 | ✓ ✓ | |
| WSC | `Question: In the sentence "{text}", does the pronoun '{span2_text}'` `refer to {span1_text}?\nAnswer: {label}` | 61 | | | 50 | ✓ | |
| RTE | `{premise}\nquestion: {hypothesis} True or False?\nanswer: {label}` | 75 | | | 85 | | |
| CB | `{premise}\nquestion: hypothesis. true, false or neither?\nanswer: {label}` | 98 | | ✓ | 95 | | ✓ |
| COPA | `Context: {premise}\nAnswer: {answer}` | 21 | | ✓ | 22 | ✓ | ✓ |
| MultiRC | `Context: {paragraph}\n{question}\n{answer}\nanswer: {label}` | 350 | ✓ | ✓ | 350 | ✓ | ✓ |
| MR | `Review: {text}\nSentiment: {label}` | 36 | | ✓ | 40 | ✓ | ✓ |
| Subj | `input: {text}\ntype: {label}` | 40 | | ✓ | 40 | ✓ | ✓ |

Table 11: Details of the datasets and prompts used for the ICL evaluation of our OPT-2.7B and Llama-2-7B AutoCompressors and baselines. "Toks / dem." (*Tokens per demonstration*) denotes how long demonstrations are for the average example. "Cal." (*Calibration*) denotes whether we use calibration (Sachan et al., 2022), and "Bal." (*Balanced*) means whether we enforce class-balanced sampling. We decide the ticks based on which method performs best on a held-out validation set.

| | AG News | SST-2 | BoolQ | WiC | WSC | RTE | CB | COPA | MultiRC | MR | Subj |
|---|---|---|---|---|---|---|---|---|---|---|---|
| | | | | | *OPT-2.7B AutoCompressor* | | | | | | |
| Zero-shot | $68.2_{(0.0)}$ | $78.0_{(0.0)}$ | $\mathbf{60.2}_{(0.0)}$ | $49.5_{(0.0)}$ | $60.6_{(0.0)}$ | $55.2_{(0.0)}$ | $43.6_{(0.0)}$ | $69.0_{(0.0)}$ | $43.8_{(0.0)}$ | $60.0_{(0.0)}$ | $56.7_{(0.0)}$ |
| 50 summary vecs. | $\mathbf{72.7}_{(1.4)}$ | $84.3_{(9.2)}$ | $55.8_{(4.2)}$ | $50.4_{(1.0)}$ | $61.3_{(5.8)}$ | $54.8_{(3.4)}$ | $55.9_{(5.4)}$ | $71.6_{(0.6)}$ | $44.1_{(1.1)}$ | $70.4_{(10.2)}$ | $\mathbf{63.2}_{(7.7)}$ |
| 100 summary vecs. | $71.2_{(3.8)}$ | $\mathbf{87.0}_{(3.5)}$ | $57.5_{(4.6)}$ | $50.7_{(1.0)}$ | $60.2_{(6.7)}$ | $55.5_{(2.5)}$ | $54.4_{(4.0)}$ | $\mathbf{71.9}_{(0.4)}$ | $45.6_{(2.8)}$ | $73.1_{(12.9)}$ | $62.2_{(5.8)}$ |
| 150 summary vecs. | $68.2_{(3.3)}$ | $82.6_{(5.6)}$ | $59.8_{(1.8)}$ | $\mathbf{51.8}_{(1.1)}$ | $\mathbf{63.5}_{(0.0)}$ | $\mathbf{55.8}_{(1.8)}$ | $\mathbf{58.3}_{(5.1)}$ | $71.4_{(0.5)}$ | $\mathbf{46.7}_{(2.1)}$ | $67.0_{(11.9)}$ | $58.5_{(6.7)}$ |
| ICL (150 tokens) | $72.5_{(2.5)}$ | $70.8_{(12.6)}$ | $60.2_{(0.0)}$ | $50.4_{(1.1)}$ | $52.3_{(13.9)}$ | $57.6_{(4.3)}$ | $51.1_{(7.1)}$ | $71.3_{(1.5)}$ | $43.8_{(0.0)}$ | $\mathbf{86.4}_{(4.2)}$ | $61.7_{(11.2)}$ |
| ICL (750 tokens) | $67.3_{(3.4)}$ | $87.5_{(5.0)}$ | $69.1_{(1.0)}$ | $51.0_{(1.7)}$ | $62.9_{(0.8)}$ | $57.4_{(4.4)}$ | $49.0_{(1.1)}$ | $72.0_{(0.7)}$ | $52.0_{(5.4)}$ | $86.7_{(5.9)}$ | $73.6_{(13.9)}$ |
| | | | | | *OPT-2.7B RMT* | | | | | | |
| Zero-shot | $66.9_{(0.0)}$ | $72.8_{(0.0)}$ | $\mathbf{58.4}_{(0.0)}$ | $50.3_{(0.0)}$ | $\mathbf{64.4}_{(0.0)}$ | $55.2_{(0.0)}$ | $42.2_{(0.0)}$ | $68.8_{(0.0)}$ | $43.9_{(0.0)}$ | $62.5_{(0.0)}$ | $\mathbf{69.8}_{(0.0)}$ |
| 1-step summary vecs. | $66.3_{(5.5)}$ | $86.5_{(5.1)}$ | $49.6_{(8.1)}$ | $51.0_{(1.00)}$ | $57.7_{(6.6)}$ | $51.3_{(1.2)}$ | $\mathbf{53.3}_{(3.8)}$ | $67.4_{(1.1)}$ | $44.9_{(1.2)}$ | $52.6_{(2.8)}$ | $63.3_{(11.2)}$ |
| 2-step summary vecs. | $65.2_{(7.2)}$ | $\mathbf{88.6}_{(2.3)}$ | $54.8_{(4.1)}$ | $50.3_{(0.8)}$ | $58.6_{(6.7)}$ | $50.2_{(1.4)}$ | $49.5_{(4.8)}$ | $68.2_{(1.2)}$ | $\mathbf{45.5}_{(1.8)}$ | $54.1_{(1.9)}$ | $54.6_{(1.7)}$ |
| 3-step summary vecs. | $63.9_{(3.3)}$ | $84.5_{(6.6)}$ | $41.8_{(9.7)}$ | $50.6_{(0.6)}$ | $54.3_{(7.9)}$ | $50.2_{(1.4)}$ | $49.5_{(3.6)}$ | $68.0_{(0.9)}$ | $45.5_{(1.0)}$ | $52.8_{(1.6)}$ | $58.4_{(8.6)}$ |
| ICL (150 tokens) | $\mathbf{70.8}_{(1.9)}$ | $75.1_{(13.3)}$ | $58.4_{(0.0)}$ | $\mathbf{51.7}_{(2.8)}$ | $52.5_{(13.1)}$ | $57.2_{(3.6)}$ | $46.5_{(3.6)}$ | $\mathbf{69.3}_{(1.5)}$ | $43.9_{(0.0)}$ | $\mathbf{89.0}_{(1.4)}$ | $60.7_{(12.1)}$ |
| ICL (750 tokens) | $65.8_{(4.2)}$ | $85.7_{(9.7)}$ | $57.2_{(7.6)}$ | $51.5_{(2.7)}$ | $59.2_{(8.5)}$ | $57.8_{(2.0)}$ | $48.2_{(0.7)}$ | $70.9_{(0.7)}$ | $54.6_{(3.6)}$ | $87.5_{(4.6)}$ | $71.6_{(12.6)}$ |
| | | | | | *OPT-2.7B Pre-trained* | | | | | | |
| Zero-shot | $65.1_{(0.0)}$ | $79.1_{(0.0)}$ | $55.8_{(0.0)}$ | $49.4_{(0.0)}$ | $53.9_{(0.0)}$ | $51.2_{(0.0)}$ | $21.2_{(0.0)}$ | $66.8_{(0.0)}$ | $43.7_{(0.0)}$ | $59.0_{(0.0)}$ | $66.2_{(0.0)}$ |
| ICL (150 tokens) | $71.6_{(2.6)}$ | $68.56_{(14.9)}$ | $55.8_{(0.0)}$ | $50.6_{(1.0)}$ | $53.30_{(11.1)}$ | $56.1_{(2.4)}$ | $46.2_{(6.4)}$ | $71.7_{(1.2)}$ | $43.7_{(0.0)}$ | $86.7_{(4.3)}$ | $61.9_{(10.9)}$ |
| ICL (750 tokens) | $63.3_{(5.1)}$ | $91.0_{(3.2)}$ | $63.0_{(1.3)}$ | $50.0_{(0.4)}$ | $63.5_{(0.6)}$ | $54.7_{(3.0)}$ | $52.1_{(4.8)}$ | $73.4_{(1.0)}$ | $53.5_{(6.2)}$ | $89.9_{(2.2)}$ | $64.4_{(10.7)}$ |
| | | | | | *Llama-2-7B AutoCompressor* | | | | | | |
| Zero-shot | $63.3_{(0.0)}$ | $67.7_{(0.0)}$ | $67.4_{(0.0)}$ | $50.8_{(0.0)}$ | $43.3_{(0.0)}$ | $58.8_{(0.0)}$ | $42.9_{(0.0)}$ | $52.5_{(0.0)}$ | $52.5_{(0.0)}$ | $57.4_{(0.0)}$ | $49.3_{(0.0)}$ |
| 50 summary vecs. | $79.6_{(4.9)}$ | $\mathbf{94.2}_{(1.6)}$ | $\mathbf{70.1}_{(3.3)}$ | $51.6_{(2.1)}$ | $47.7_{(8.7)}$ | $66.3_{(7.0)}$ | $46.4_{(18.7)}$ | $84.5_{(1.0)}$ | $52.6_{(2.8)}$ | $\mathbf{91.5}_{(1.0)}$ | $53.5_{(3.6)}$ |
| 100 summary vecs. | $\mathbf{87.6}_{(1.2)}$ | $92.6_{(3.3)}$ | $66.3_{(2.8)}$ | $52.5_{(2.2)}$ | $42.9_{(2.5)}$ | $63.5_{(6.6)}$ | $\mathbf{64.5}_{(5.9)}$ | $85.9_{(0.4)}$ | $\mathbf{56.1}_{(1.2)}$ | $90.7_{(2.6)}$ | $\mathbf{57.0}_{(5.6)}$ |
| 150 summary vecs. | $85.4_{(3.4)}$ | $92.3_{(2.9)}$ | $68.0_{(1.8)}$ | $\mathbf{52.8}_{(1.5)}$ | $49.9_{(7.6)}$ | $65.3_{(6.6)}$ | $54.8_{(5.8)}$ | $\mathbf{86.1}_{(0.6)}$ | $54.8_{(2.2)}$ | $91.1_{(2.2)}$ | $56.6_{(7.9)}$ |
| ICL (150 tokens) | $74.5_{(2.2)}$ | $92.4_{(3.1)}$ | $67.4_{(0.0)}$ | $52.4_{(2.7)}$ | $\mathbf{51.8}_{(6.9)}$ | $\mathbf{69.1}_{(2.1)}$ | $46.4_{(23.0)}$ | $80.0_{(1.9)}$ | $52.5_{(0.0)}$ | $79.7_{(15.7)}$ | $57.9_{(10.7)}$ |
| ICL (750 tokens) | $81.2_{(4.1)}$ | $93.8_{(1.2)}$ | $67.7_{(2.7)}$ | $52.4_{(2.0)}$ | $40.0_{(5.7)}$ | $73.1_{(3.5)}$ | $50.3_{(2.8)}$ | $82.6_{(1.6)}$ | $47.0_{(3.2)}$ | $91.6_{(0.8)}$ | $60.7_{(14.8)}$ |
| | | | | | *Llama-2-7B Pre-trained* | | | | | | |
| Zero-shot | $68.8_{(0.0)}$ | $87.2_{(0.0)}$ | $70.0_{(0.0)}$ | $51.4_{(0.0)}$ | $65.4_{(0.0)}$ | $62.8_{(0.0)}$ | $32.1_{(0.0)}$ | $75.5_{(0.0)}$ | $54.5_{(0.0)}$ | $84.1_{(0.0)}$ | $48.9_{(0.0)}$ |
| ICL (150 tokens) | $71.9_{(3.8)}$ | $91.6_{(2.9)}$ | $70.0_{(0.0)}$ | $51.0_{(1.9)}$ | $55.4_{(3.2)}$ | $70.9_{(1.7)}$ | $39.3_{(21.2)}$ | $84.2_{(1.3)}$ | $54.5_{(0.0)}$ | $90.6_{(3.3)}$ | $63.6_{(10.8)}$ |
| ICL (750 tokens) | $78.2_{(3.8)}$ | $94.5_{(0.8)}$ | $70.3_{(6.1)}$ | $54.9_{(1.9)}$ | $42.2_{(5.0)}$ | $71.3_{(4.4)}$ | $51.3_{(3.5)}$ | $85.3_{(0.7)}$ | $47.0_{(1.5)}$ | $92.9_{(0.5)}$ | $65.4_{(14.5)}$ |

Table 12: We evaluate the following models on 11 in-context learning tasks: The OPT-2.7B AutoCompressor and RMT model, the Llama-2-7B AutoCompressor, and the respective pre-trained models. For each fine-tuned model, numbers in bold are the highest evaluation results using at most 150 additional tokens. When using summary vectors, the OPT-2.7B AutoCompressor outperforms the RMT model on 8/11 tasks. Moreover, the OPT-2.7B AutoCompressor benefits from multiple compression steps on most tasks whereas the RMT model performs best without summary vectors on 7/11 tasks and benefits from 3-step summary vectors on none of the above tasks. The Llama-2 AutoCompressor achieves the absolute highest accuracy using summary vectors on 7/11 tasks. It also achieves the highest accuracy with summary vectors on 9/11 tasks using at most 150 additional tokens.