# OpenReview forum: "Adapting Language Models to Compress Contexts"
_EMNLP/2023/Conference — EMNLP 2023 Main_

### Official Review · Reviewer_PB37 · 2023-08-03

**Soundness:** 4

**Excitement:**

3: Ambivalent: It has merits (e.g., it reports state-of-the-art results, the idea is nice), but there are key weaknesses (e.g., it describes incremental work), and it can significantly benefit from another round of revision. However, I won't object to accepting it if my co-reviewers champion it.

**Paper Topic And Main Contributions:**

The author proposes teaching pre-trained LMs the skill to condense text into summary vectors, which are concise soft prompts. The models they introduce, named AutoCompressors, are developed using an uncomplicated unsupervised training strategy. This strategy motivates the model to retain vital data within the summary vectors. The author introduces a method called summary accumulation, where vectors from all segments come together to form the summary for the whole document. They also train AutoCompressors on inputs split up at random. This ensures the models are more effective in condensing texts of various lengths for different tasks. The analysis implies that summary vectors excel in specific tasks, outdoing few-shot ICL in 8 of 11 evaluated tasks. AutoCompressors were also explored for two applications: retrieval-augmented language modeling, where summary vectors yielded 1.5 times better comprehension than standard texts, and zero-shot passage re-ranking, where they provided the best balance of performance and speed.

**Reasons To Accept:**

-The results show that all modules are helpful
-method is efficient and requires less memory for training
-This work is clearly written and easy to follow.
-the applications are pratical

**Reasons To Reject:**

-innovativeness isn't high
-background of RMT is not discussed in main context

**Reproducibility:**

4: Could mostly reproduce the results, but there may be some variation because of sample variance or minor variations in their interpretation of the protocol or method.

**Reviewer Confidence:**

3: Pretty sure, but there's a chance I missed something. Although I have a good feel for this area in general, I did not carefully check the paper's details, e.g., the math, experimental design, or novelty.

---

> ### Author Rebuttal · Authors · 2023-08-28
>
> Thank you for your review! We are pleased to see that the reviewer appreciates the thoroughness of our evaluation process and notes the exciting practical implications of our work, which shows that concise summary vectors are more information-dense than plain text for a wide variety of applications. These soft prompt-based applications open up exciting new research avenues, and we discuss below how our AutoCompressors constitute an important new contribution compared to previous works.
>
> $~$
> > Limited innovativeness
>
> We respectfully disagree with this point. While our approach indeed shares similarities with RMT (which we stated in both the introduction and methodology), our work introduced new techniques in methodology including
> 1. Summary accumulation,  which summarizes vectors from all previous segments
> 2. Randomized segmenting, which allows better compressing texts of variable lengths in downstream tasks.
> 3. Stop-gradients, which reduces computational resources required for fine-tuning
> More importantly, these innovations improve long-range language modeling performance consistently and enable new applications of summary vectors in downstream tasks (e.g., retrieval-augmented settings), as we demonstrated in our experiments.
>
> Compared to the RMT paper, we demonstrate the possibility of adapting pre-trained large language models (up to 2.7B parameters; we also plan to include results of LLaMA-7B models in our final version; see our response to Db2M if interested), and we show novel applications in downstream tasks, including a) in-context learning with many examples; b) retrieval-augmented language modeling, and c) unsupervised passage re-ranking. By contrast, the RMT paper trained small models from scratch and only evaluated them on language modeling and synthetic tasks.  Hence, we believe that our approach is novel and widely applicable in many scenarios.
>
> $~$
> > RMT not discussed in main context
>
> We are surprised that the reviewer feels that the connection to the RMT work was not drawn out enough in our main text. RMT is discussed in the introduction (Line 061-072), and we recap the RMT architecture in our main sections 3.1 and 3.2, where we go over the key differences between AutoCompressors and RMT models, including summary accumulation, randomized segmenting, and stop-gradients during fine-tuning. Moreover, we extensively benchmark our AutoCompressors against RMT. We would appreciate more detailed feedback on how we can improve the writing and further highlight the connection to RMT.

---

### Official Review · Reviewer_9A3j · 2023-08-05

**Soundness:** 4

**Excitement:**

4: Strong: This paper deepens the understanding of some phenomenon or lowers the barriers to an existing research direction.

**Paper Topic And Main Contributions:**

Authors attempt to tackle the problem of processing long documents by proposing a novel LLM that can compress long contexts into “summary vectors”. Their method replicates the principle of the Recurrent Memory Transformer architecture, with some differences (like the fact that they accumulate summary vectors from segment to another). Their model processes long documents by fragmenting them to segments, prepend previously accumulated summary vector to each segment before recurrently processing it. At the end, the model outputs the representation of the whole document plus a small (compressed) vector that summarizes all the segments (the whole document). Authors demonstrate how such segments could be used to improve models’ performance in many tasks, such as in in-context-learning, retrieval-based and passage reranking tasks. The AutoCompressor (authors’ model) needs to be initialized by a pretrained model (OPT for instance) and to trained to produce summary vectors (through a language modeling objective). There is no finetuning on downstream tasks. Authors did a remarkable work in heavily experimenting their model, though it is not completely flawless.

**Reasons To Accept:**

1)	Well written.
2)	Solid experimental setting.
3)	Authors methodology is well argumented.

**Reasons To Reject:**

1)	I think authors missed a relevant benchmark (Long range arena).

**Reproducibility:**

3: Could reproduce the results with some difficulty. The settings of parameters are underspecified or subjectively determined; the training/evaluation data are not widely available.

**Reviewer Confidence:**

3: Pretty sure, but there's a chance I missed something. Although I have a good feel for this area in general, I did not carefully check the paper's details, e.g., the math, experimental design, or novelty.

---

> ### Author Rebuttal · Authors · 2023-08-28
>
> Thank you for your review! We are glad to see that you find our new applications interesting, and that you appreciate the effort we put into checking every aspect of our method through careful ablations and benchmarking.
>
>
> We are also happy that you note that we do not fine-tune on any downstream tasks and that our positive results use a single unsupervised fine-tuning objective. As a result, there is substantial follow-up work to investigate how AutoCompressors can be further fine-tuned on specific downstream applications. The unsupervised nature of our work is an important feature which we chose to respect in all our evaluations. It is exciting that AutoCompressors can reason over summary vectors obtained from out-of-domain text and that pre-trained summary vectors support new operations, like Fused Summaries.
>
> $~$
> > Missed a relevant benchmark - Long Range Arena
>
> Thank you for pointing out another interesting evaluation setting for efficient long-range transformers. However, we would argue that evaluating on Long Range Arena (LRA) is beyond the scope of the current paper for several reasons.
>
> Firstly, most tasks in LRA do not focus on text data, and are not suitable for adapting pre-trained language models, as they include image classification tasks, the Pathfinder tasks, and the synthetic ListOps tasks. Secondly, due to the diversity of tasks, LRA requires models to be trained for each task individually, and we feel that this is a departure from the unsupervised finetuning approach in many recent long-range LM works. LRA is a good benchmark to understand the architectural implications of AutoCompressors trained from scratch, but in this paper we focus on natural language applications, such as document retrieval and re-ranking.

---

### Official Review · Reviewer_Db2M · 2023-08-05

**Soundness:** 4

**Excitement:**

4: Strong: This paper deepens the understanding of some phenomenon or lowers the barriers to an existing research direction.

**Paper Topic And Main Contributions:**

The work proposes a new method to tackle the long input-length context problems with language models. It adapts pre-trained LMs into AutoCompressors i.e. compressing long contexts into compact summary vectors as soft prompts. The idea is simple yet effective. In my perspective, the more interesting part is to maintain the low computational overhead while caching the summaries both during training (computational graph) as well as inference (caching).

**Questions For The Authors:**

1) Line 213-218 (Positional embeddings), the argument is a little unconvincing. Is it the case with all the types of position embeddings such as RoPE?
2) It seems the position of summary tokens is not preserved (by excluding positional information), it would be interesting to see the impact of positional embedding even if it has limitations such as scalability.
3) It would be great to see the results for other architectures and maybe other than (causal) decoder-only settings.

**Reasons To Accept:**

There are various reasons that make me excited about the approach:
1) Summary vectors soft prompts are 1-2 orders of magnitude lower than plain text which has not been achieved before.
2) One can extend the context window significantly with minimal computational overhead.
3) The idea emerges to be able to reason over summary vectors.
4) It can help reduce the space occupied by ICL examples (demonstrations) and definitions.

Overall, the paper is well-written and the contributions are strong.

**Reasons To Reject:**

In particular, I don't have any strong reason to reject the work.  I have included my questions in the question for the authors section.

**Reproducibility:**

5: Could easily reproduce the results.

**Reviewer Confidence:**

4: Quite sure. I tried to check the important points carefully. It's unlikely, though conceivable, that I missed something that should affect my ratings.

---

> ### Author Rebuttal · Authors · 2023-08-28
>
> Thank you for your review and very interesting and pertinent questions! We are pleased to see that you appreciate the simplicity of our method, and we agree that one of the main advantages of AutoCompressors lies in the ability to cache compressed corpora to speed up inference over frequently seen contexts. Please find some comments below addressing your questions about positional embeddings and alternative architectures..
>
> $~$
> > AutoCompressors with other type of positional embeddings such as RoPE
>
> This is a fascinating question! Indeed, we have done follow-up experiments for LLaMA models with RoPE embeddings, and find that for RoPE embeddings, keeping the normal relative embeddings for summary tokens performs the best. In contrast to absolute position embeddings, this also will not reduce the context window nor introduce newly initialized parameters. We will add the LLaMA results in the final version.
>
> $~$
> > Positional information between different summary tokens is not preserved
>
> Thank you for raising this point and acknowledging the limitations this would have on scalability. In small-scale ablation experiments, we find that adding explicit positional embeddings to the summary tokens makes no discernable impact on performance. This is also in agreement with [1], which shows that Transformers with causal masking can easily reconstruct useful positional information.
>
> $~$
> >It would be great to see the results for other architectures and maybe other than (causal) decoder-only settings.
>
> In follow-up efforts to scale up to larger models and to adapt AutoCompressors to other architectures, we have successfully fine-tuned AutoCompressors from LLaMA-7b models. We will include the results in the final version and release the models publicly. We are pleased to report that AutoCompressors are more effective for LLaMA models than OPT models: fine-tuning on the same corpus and conditioning on 6144 compressed tokens gives perplexity gains of 8.1% in-domain and 10.4% out-of-domain over a fine-tuned LLaMA model (compared to gains of 5.6% and 5.0% for OPT 2.7b-based AutoCompressors reported in Table 1).
> We are also very interested in applying AutoCompressors to other architectures, such as encoder-decoder, and are looking forward to pursuing this in future work.
>
> $~$
> [1] Kazemnejad, et al., 2023. The Impact of Positional Encoding on Length Generalization in Transformers

---

### Meta-Review · Area_Chair_Pq59 · 2023-09-28

**Recommendation:** 5

**Metareview:**

All reviewers agree on the soundness and excitement of this paper. The reviewers specifically call out the significance of the result by which sequence length in LLMs can be extended to much larger while also speeding up inference. The paper is well written with solid experimental results. The community will benefit from this work being published at EMNLP.

---

### Decision · Program_Chairs · 2023-10-07

**Decision:**

Accept-Main

**Comment:**

All reviewers agree on the soundness and excitement of this paper. The reviewers specifically call out the significance of the result by which sequence length in LLMs can be extended to much larger while also speeding up inference. The paper is well written with solid experimental results. The community will benefit from this work being published at EMNLP.